

# Time perception and lived experience in personality disorders: differences across types, dimensions and severity

Anna Sterna[1], Eisuke Sakakibara[2] and Marcin Moskalewicz[1,3]

[1] Philosophy of Mental Health Unit, Department of Social Sciences and the Humanities, Poznań University of Medical Sciences, Poznań, Poland
[2] Department of Neuropsychiatry, Graduate School of Medicine, University of Tokyo, Tokyo, Japan
[3] Institute of Philosophy, Maria Curie-Sklodowska University Lublin, Lublin, Poland

Corresponding author
Anna Sterna, an.sterna@gmail.com

## ABSTRACT

**Background:** Altered temporal experience lies at the core of various psychiatric conditions, including borderline personality disorder (BPD). Mainstream research in psychopathology tends to explore BPD with scrutiny while neglecting other personality disorders (PD). At the same time, the dimensional approach to PD proposes looking through the disorders' subtypes and tracing lived experience-based commonalities. This study is the first to explore the temporality of PD by investigating the relationship between symptom severity and lived time and combining objectified measures of time perception with phenomenological interpretation.

**Methods:** A total of 63 participants of various educational backgrounds, with personality disorders (36.5% male), following ICD-10 coding diagnosed with paranoid (3.2%), borderline (41.3%), narcissistic (33.3%), avoidant (4.8%), dependent (1.6%) and unspecified (15.9%) personality disorder. Levels of personality functioning and intensity of maladaptive trait domains were controlled with Level of Personality Functioning—Brief Scale 2.0 and Personality Inventory for ICD-11, respectively, resulting in the overall sample classification as comprising nine subclinical, 13 mild, 20 moderate, 16 severe, and five extremely severe conditions. Polish Short Version of the Zimbardo Time Perspective Inventory (PS-ZTPI) and Cottle's Circles Test (CT) were used to assess the temporal experience.

**Results:** In comparison to healthy individuals, those with PD are more oriented toward past negative (4.01 *vs.* 2.98) and less toward past positive (2.31 *vs.* 3.71) and future (3.04 *vs.* 3.47), as measured with PS-ZTPI; their pre-reflective temporal experience, as measured with CT, is dominated either by the past or the future, while the present remains marginalized. BPD distinctiveness among other PD lies in higher orientation toward hedonistic present and lower orientation toward the future. While the general temporal profile of PD is independent of age and duration of hospitalization, it is related to the severity of the condition. The more severe the impairments in self-functioning, the higher the negative past perspective and pre-reflective past dominance, and the lower the positive and future perspective. The results of this study highlight temporality as an essential aspect of lived experience in PD, being possibly related to disturbed self-experience.

# INTRODUCTION

## Alterations of lived time in mental illness and borderline personality

Different psychiatric conditions are often phenomenologically associated with disruptions of lived time. This includes mania (*Martin, Gergel & Owen, 2019*), depression (*Gallagher, 2012*), addictions (*Messas, 2014*; *Moskalewicz & Messas, 2022*), schizophrenia (*Fuchs & Pallagrosi, 2018*), autism (*Nilsson et al., 2019*) and borderline personality disorder (BPD) (*Fuchs, 2007*; *Lo Monte & Englebert, 2018*; *Stanghellini & Mancini, 2018*). Phenomenological literature theorizes the temporal experience of individuals diagnosed with BPD as *immediate* (*Fuchs, 2007*) and *instantaneous* (*Stanghellini & Mancini, 2018*); however, the research is mostly restricted to theoretical insights, with scarce empirical studies. The exceptions include research on the experienced passage of time in BPD as fragmented and discontinuous (*Sterna & Moskalewicz, 2022*) and BPD temporality being dominated by the present (*Mancini & Stanghellini, 2020*). Qualitative research additionally revealed that in BPD past is negatively biased and disorganized (*Jørgensen et al., 2012*; *Rasmussen et al., 2017*; *Spodenkiewicz et al., 2013*). However, given the limited, empirical research on BPD lived time and lack of the relevant research on other PD's temporality, it is still largely unknown what is the exact characteristics of felt time in this group.

Temporal experience may be conceptualized and operationalized in many different ways. One approach is to track the so-called time perspective (*Zimbardo & Boyd, 1999*). Time perspectives are individual or collective cognitive frames regarding past, present, and future, which may take part in decision-making and actions and do not refer to other aspects of subjective temporal experience such as sense of duration or felt tempo. The overall time perspectives profile also reflects one's ability to migrate between various temporal horizons (*Stolarski et al., 2020*). The Zimbardo Time Perspective Inventory (ZTPI) operationalizes time perspectives *via* five factors: past-negative, past-positive, present-hedonistic, present-fatalistic, and future perspectives, and it was widely used to explore different clinical samples (*McKay et al., 2018*; *Van Beek et al., 2011*). For example, patients with depression had a more negative past perspective and less hedonistic and more fatalistic views of the present compared with non-depressed subjects (*Lefèvre et al., 2019*). Recent literature also showed that an overall, balanced time perspective is correlated with subjective well-being or level of cognitive resources (*Olivera-Figueroa et al., 2023*; *Stolarski, Bitner & Zimbardo, 2011*; *Stolarski, Wiberg & Osin, 2014*; *Stolarski et al., 2020*). Among patients with schizophrenia, an unbalanced time perspective was associated with severe symptom severity (*Damiani et al., 2023*). The relation between time perspectives and various psychopathological conditions led to emergence of a distinct psychotherapeutic modality, namely Time Perspectives Psychotherapy (*Sword, Sword & Brunskill, 2014*; *Zimbardo & Sword, 2017*), applied in *e.g.*, PTSD (*Zimbardo, Sword & Sword, 2012*). In the realm of personality pathology, research using ZTPI is limited to

borderline experience. It indicates a higher orientation toward past negative and lower towards past positive perspectives in comparison to healthy controls ($N$ = 17 + 17) (*Mioni et al., 2020*). It was also shown that individuals with BPD exhibited higher present hedonistic, higher present fatalistic, and lower past positive time perspective in comparison to obsessive compulsive disorder ($N$ = 28 + 28) (*Stefanatou et al., 2016*). The only study including a broader sample of cluster B of PD ($N$ = 25) indicated lower past positive scores than healthy controls (*Oyanadel & Buela-Casal, 2014*). To date, however, no study of time perspective has investigated PDs outside cluster B diagnosis nor examined the alteration of time *via* the dimensional model of personality disorders.

Another complementary approach to objectifying time perspective is Cottle's Circles Test (*Cottle, 1967*). It measures the relatedness and balance between the sense of past, present, and future *via* a projective instrument. Although the tool has been rarely used in clinical samples over the past decades, it was recently applied in research on patients with cancer (*Van Laarhoven et al., 2011*) and with brain tumor (*Shigemune et al., 2021*) and is widely used in healthy samples (*Belovol, Boyko & Shurupova, 2021*; *Mello, Finan & Worrell, 2013*; *Wiberg et al., 2017*). To date, however, it has not been administered to patients with psychiatric conditions. Being a projective tool, Circles Test has limitations; nevertheless, it is still useful as an auxiliary instrument that may help to contextualize findings from the established questionnaires, such as ZTPI.

### From categorical to dimensional approach to personality disorders in research on temporal experience

Unfortunately, research in personality disorders (PD) has predominantly focused on BPD both in terms of symptomatology and lived experience, leaving other PD types underexplored. The only contributions to lived temporality research in PD other than BPD concerned narcissistic personality disorder (NPD). Narcissism has been conceptualized as involving timelessness (*Boschan, 1990*) or "murdering time" (*Green, 2007*) and has been associated with a negative view of one's past (*Zajenkowski et al., 2021*). It is a general trend, not limited to lived time exploration, that research on personality disorders has not advanced outside of some PD categories (*Blashfield & Intoccia, 2000*). At the same time, the categorical diagnosis of PD has been criticized for making up multiple comorbidities among PD, leaving a large number of patients outside any specific PD category, and allowing significant heterogeneity within the same category (*Widiger & Trull, 2007*). For those reasons, the alternative model of personality disorders (AMPD) in Section III of the 5th edition of the Diagnostic and Statistical Manual of Mental Disorders (DSM-5) and its Text Revision (DSM-5-TR) as well as the 11th edition of International Classification of Diseases (ICD-11) adopted dimensional approach to personality disorders (*American Psychiatric Association, 2013*, *2022*; *World Health Organization, 2022*). The dimensional model shifts the psychopathological focus from the specificity and distinctiveness of various PD types to the issue of transdiagnostic commonality and its continuous variation. In this context, we recently proposed the first phenomenological conceptualization of lived time across PD (*Sterna et al., 2024*). We suggested that individuals with PD are trapped in the past-present loops, which, along with a diminished sense of futurity, may reflect the

experiential core of general diagnostic criteria for PD, namely inflexibility and pervasiveness of maladaptive patterns (*American Psychiatric Association, 2022*).

In line with this approach, the current research aims to explore the temporal experience of individuals diagnosed with various PD categories with subclinical to extreme severity of symptomatology. Although our theoretical background lies in phenomenological psychopathology, this paper extends the typical scope of phenomenology of time consciousness towards objectified quantitative measures. The goal is to compare measurable time perspectives across the classical PD categories and to investigate how temporal experience relates to the severity of PD and to maladaptive personality traits according to the dimensional model. Finally, the research intends to present the quantitative results against the phenomenological insights on atypical temporality in PD and thus contribute to bridging empirical research results with philosophical (phenomenological) discourse.

## MATERIALS AND METHODS

### Sample recruitment procedure and characteristics

Sixty-three participants with personality disorders were recruited from a larger group of in-patients admitted in 2023 to Neurosis and Personality Disorders Ward in Psychiatric Hospital Międzyrzecz in Poland. The enrollment procedure comprised several stages: (a) we initially identified 120 patients, who were all diagnosed with PD by at least three skilled psychiatrist (PD diagnosis first established in outpatient care), and then confirmed by two psychiatrists working in the ward (with 7 and 15 years of professional experience respectively) and followed by clinical psychologist's assessment (along six consultation sessions); (b) next, we applied the following exclusion criteria: suicidal crisis or psychotic decompensation and comorbid substance use disorders; this reduced the initial sample to 100 patients; (c) next, we did intensity sampling so information-rich but not extreme cases were enrolled to increase the comprehensiveness of the studied phenomena (*Benoot, Hannes & Bilsen, 2016*) resulting in final sample of 63 participants. Step (c) was processed by the first author, who is a skilled clinical psychologist with 10 years of professional experience in PD diagnosis, and who further confirmed the validity of the diagnosis *via* clinical documentation analysis and clinical consensus with Ward's team when needed. This procedure and selection criteria were chosen to assure the diagnostic accuracy of the study sample. All participants signed an informed consent in a paper format. The study protocol was approved by the hospital and Poznań University of Medical Science's bioethical committee (decision no: KB-367/23). Portions of this text were previously published as part of a preprint (*Sterna, Sakakibara & Moskalewicz, 2024*).

Among 63 participants, 63.5% ($N = 40$) were female, aged from 18 to 64 years (mean = 28.5, SD = 10.5). According to the ICD-10 coding, 3.2% ($N = 2$) were diagnosed with paranoid (PPD), 41.3% ($N = 26$) with borderline (BPD), 33.3% ($N = 21$) with narcissistic (NPD), 4.8% ($N = 3$) with avoidant (AvPD), 1.6% ($N = 1$) with dependent (DPD), and 15.9% ($N = 10$) with unspecified (UPD). Hospitalization lasted from 2 to 20 weeks, with a mean duration of 7.1 weeks (mean = 6.0, SD = 4.4). Participants were of various educational backgrounds: 3.2 % ($N = 2$) had primary school education, 12.7%

($N$ = 8) had vocational training, 49.2% ($N$ = 31) had high school diploma, 14.3% ($N$ = 9) had incomplete higher education, and 20.1% ($N$ = 13) had a higher education degree.

## Measures

All the participants were tested using printed questionnaires; the procedure took approximately 30 min to complete. The authors have permission to use all the instruments from the copyright holders.

**Level of Personality Functioning Scale—Brief Form 2.0 (LPFS-BF 2.0)**—a self-report measure first introduced in section III of the DSM-5 to examine the level of personality functioning (criterion A) and then updated (*Weekers, Hutsebaut & Kamphuis, 2018*). We used Polish adaptation (*Łakuta et al., 2023*) consisting of 12 items, each assessed on a four-point Likert scale (1 = very rarely/completely untrue, 4 = very often/completely true). Two constituents of the general criterion are assessed: the degree of impairment in self- and interpersonal functioning (6–24 points each). The items are designed to capture "essential commonalities" (*Weekers, Hutsebaut & Kamphuis, 2018*, pp.1) for PD as a category, *e.g.*, identity disturbances (item example: "I often do not know who I really am"), self-directedness (item example: "I have no sense of where I want to go in my life"), and capacity for establishing intimacy (item example: "I often feel very vulnerable when relationships become more personal"). An overall result is psychometrically normalized and can be used to define the severity of PD (*Weekers et al., 2023*) and Polish adaptation was tested for reliability ($\alpha$ = 0.64) (*Łakuta et al., 2023*). We used both: raw, overall scores and cut-off scores suggested by Weekers and colleagues (12–25 "subclinical," 26–30 "mild," 31–35 "moderate," 36–40 "severe," 41–48 "extreme").

**Personality Inventory for ICD-11 (PICD)**—a self-report questionnaire designed to measure five maladaptive domains in line with the dimensional model of diagnosis proposed in ICD-11, including: negative affective, detachment, dissocial, disinhibition, and anankastic. We used Polish adaptation (*Cieciuch et al., 2022*), tested for reliability (subscales reliability range $\alpha$ = 0.77–0.82). It consists of 60 items, each assessed on a 5-point Likert scale (1 = strongly disagree, 5 = strongly agree), providing each dimension with a score range of 12–60 points. Each of the domains is composed of several personality traits, *e.g.*, *negative affective* domain comprises overreactiveness to minor stressors with negative affect including sadness, anger, anxiety *etc.* (item example: "I am usually an anxious person"), while *disinhibition* domain covers not only impulsivity but also distractibility (*Oltmanns & Widiger, 2018*) (item example: "I tend to act impulsively.").

**Polish Short Version of the Zimbardo Time Perspective Inventory (PS-ZTPI)**—a self-report measure aimed at assessing four kinds of reflective time perspective: past-negative, past-positive, present-hedonistic, and future (*Przepiorka, Sobol-Kwapinska & Jankowski, 2016*). The instrument was tested for reliability (subscales reliability range $\alpha$ = 0.69–0.78). Note that PS-ZTPI omits items for the present-fatalistic perspective scale, which was contained in the original version developed by *Zimbardo & Boyd (1999)* due to its low reliability (*Przepiorka, Sobol-Kwapinska & Jankowski, 2016*). Although the use of short version is currently debated (*McKay et al., 2015*; *Temple et al., 2017*), it is the only version adapted to Polish; being short, it also secures comfort of the procedure. PS-ZTPI

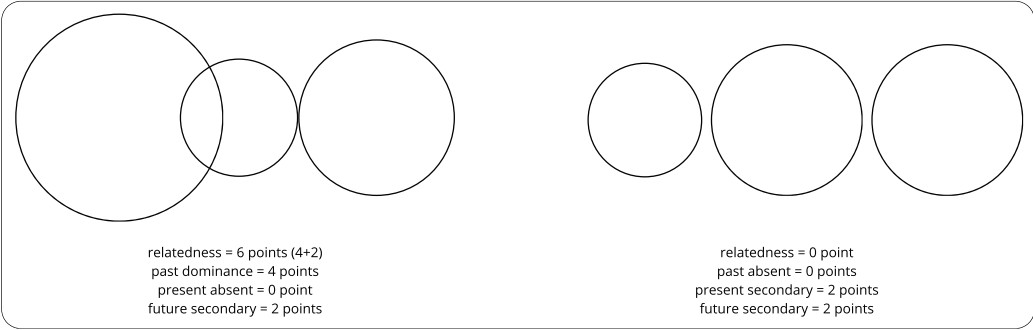

**Figure 1** Examples of drawing and scoring of T. Cottle's circles test.

consists of 20 items, each assessed on a five-point Likert scale (1 = very untrue, 5 = very true). The scores are averaged over five questions for each construct. Past negative scale is designed to reflect one's negative, full-of-regret attitude towards one's past (item example: "I often think what I should have done differently in my life."); conversely—the past positive scale measures sentimental attitude (item example: "It gives me pleasure to think about my past."). Present hedonistic scale focuses on pleasure and riskiness (item example: "I do things impulsively."), while future scale represents goal orientation (item example: "When I want to achieve something, I set goals and consider specific means for reaching those goals.").

**T. Cottle's Circles Test**—a projective tool designed to uncover pre-reflective, felt sense of temporal relatedness or separateness between temporal dimensions (*Cottle, 1967*). Participants are asked to draw three circles representing their past, present, and future. The extent to which the drawn circles overlap is interpreted as the degree of relatedness between the dimensions of time experienced by the respondents, while their size is an indicator of their potential dominance. Relatedness is scored from 0 to 18 points (two points for each of two circles being tangent to each other, four points for each overlapped region regardless of the degree of overlap, and six points if one region is fully immersed in another), and dominance is scored from 0 to 4 points (0 points—"absent," when a circle is not larger than other two circles, two points—"secondary," when a circle is larger than one other circle, and four points—"dominance," when a circle is larger than other two circles). Figure 1 shows two examples of possible drawings and their scoring.

## Statistical analysis

We conducted the analysis with the use of IBM SPSS Statistics v. 29 software. Normality of the sample was tested using Shapiro-Wilk test. Data was tested for normal distribution; whenever variables were not normally distributed we used either Kruskall-Wallis H or Mann-Whitney U test; while normally distributed variables were tested using Student's T test. The exact tests used for specific variables are specified by the numeric results provided in the tables (Temporal experience in personality disorders). Correlations were examined using Spearman's ρ. The alpha error rate was set as 0.05. No correction for multiple

comparisons was made. We interpreted 0.2–0.4 effect size as "weak," 0.4–0.7 as "moderate," and >0.7 as "strong."

## RESULTS

### Symptoms characteristics

The severity of personality dysfunction as measured with LPFS-BF 2.0 and maladaptive personality traits as assessed with PICD are presented in Table 1 (mean ± SD). Based on the overall severity score of LPFS-BF 2.0, nine patients were classified as subclinical, 13 as mild, 20 as moderate, 16 as severe, and five as extremely severe. No significant sex differences in the severity of personality dysfunction or maladaptive personality traits were observed.

### Temporal experience in personality disorders

*Time perspectives.* The first column of Table 2 show the distribution of the time perspectives subscales. We bolded the results which significantly differentiate PD sample from healthy subjects, which is specified in the discussion section. Mean ± SD of the scores of time perspectives as measured with ZTPI were 4.01 ± 0.68 for past negative, 2.31 ± 0.81 for past positive, 2.01 ± 1.00 for present hedonism, and 3.04 ± 0.85 for future perspectives. Individuals with PD were primarily oriented towards past negative staying less oriented towards present and future perspectives.

*Temporal dominance.* Figure 2 shows the distribution of temporal dominance in the PD group. The temporal experience was dominated by the past and future, while the present was marginalized. Results in bold in Table 2 are based on the interval scoring and indicate the significant differences between the PD group and Cottle's healthy sample, which are specified in discussion section.

*Temporal relatedness.* The mean relatedness score was 2.44 ± 4.05 in our PD sample. Using a categorical classification, 73% of PD participants sketched their temporal passage atomistically; their past, present, and future circles were completely isolated from each other.

*Temporal experience and linear time.* We checked if there is a relationship between time perspective as measured with ZTPI, Circles Test, and measures of objective time, namely, age and duration of hospitalization using Spearman's correlation. The results indicated that older participants scored lower on present hedonistic ($\rho = -0.314$, $p = 0.012$) and past negative ($\rho = -0.314$, $p = 0.012$) perspectives. Other correlations were statistically insignificant.

*Temporal experience and sex.* None of the Circles Test scores were sensitive to sex differences. The only difference emerged in orientation towards past negative as measured with ZTPI—women tend to be more oriented towards past negative than men (4.17 ± 0.57 *vs.* 3.73 ± 0.77, U = 315.5, $p = 0.04$).

### Differences in temporal experience across PD types

We performed group comparisons with the use of nonparametric tests. The summary of the results is presented in Table 2. There was a significant difference in orientation toward

**Table 1 Severity of personality dysfunction and maladaptive personality traits among the participants.**

| | LPFS-BF 2.0 | | | | PICD | | | |
|---|---|---|---|---|---|---|---|---|
| | Impairment in self functioning | Interpersonal disturbance | Overall severity score | Negative affective | Disinhibition | Detachment | Dissocial | Anankastic |
| All PD (N = 63) | 17.6 ± 3.9 | 14.7 ± 3.7 | 32.3 ± 6.1 | 50.8 ± 8.5 | 33.5 ± 9.9 | 32.8 ± 9.2 | 26.5 ± 7.1 | 37.5 ± 9.0 |
| Male (N = 23) | 16.8 ± 3.8 | 14.7 ± 3.9 | 31.0 ± 5.7 | 48.1 ± 8.5 | 32.8 ± 10.9 | 33.0 ± 7.9 | 28.1 ± 6.9 | 37.8 ± 9.0 |
| Female (N = 40) | 18.5 ± 3.9 | 14.7 ± 3.6 | 33.0 ± 6.2 | 52.3 ± 8.2 | 33.9 ± 9.3 | 32.7 ± 9.9 | 25.5 ± 7.2 | 37.3 ± 9.0 |

**Table 2 Differences in temporal experience among different personality disorders categories (N = 63).**

| | | All PD (N = 63) | BPD (N = 26) | NPD (N = 21) | Other PD (N = 16) | BPD vs. NPD | BPD vs. other PD |
|---|---|---|---|---|---|---|---|
| ZTPI | Past negative | **4.01 ± 0.68** | 4.24 ± 0.65 | 3.76 ± 0.67 | 3.98 ± 0.63 | **$p = 0.008$[U]** | $p = 0.13$[H] |
| | Past positive | **2.31 ± 0.81** | 2.11± 0.71 | 2.43 ± 0.82 | 2.48 ± 0.90 | $p = 0.192$[U] | $p = 0.78$[H] |
| | Present hedonistic | 3.01 ± 1.00 | 3.57± 0.89 | 2.80 ± 0.95 | 2.39 ± 0.77 | **$p = 0.003$[T]** | **$p < 0.001$[H]** |
| | Future | **3.04 ± 0.85** | 2.83 ± 0.67 | 3.00 ± 0.86 | 3.45 ± 0.99 | $p = 0.18$[U] | **$p = 0.01$[H]** |
| CT | Past dominance | **2.00 ± 1.72** | 2.23 ± 1.63 | 1.71 ± 1.82 | 1.63 ± 1.67 | $p = 0.15$[U] | $p = 0.13$[H] |
| | Present dominance | 1.75 ± 1.37 | 1.77 ± 1.32 | 1.43 ± 1.43 | 1.87 ± 1.36 | $p = 0.21$[U] | $p = 0.91$[H] |
| | Future dominance | **1.75 ± 1.81** | 1.77 ± 1.82 | 1.71 ± 1.82 | 2.13 ±1.86 | $p = 0.73$[U] | $p = 0.31$[H] |
| | Relatedness | 2.44 ± 4.05 | 1.62 ± 3.92 | 3.24 ± 4.48 | 1.63 ± 3.67 | $p = 0.20$[U] | $p = 0.62$[H] |
| | Atomistic perspective | 73% | 80.8% | 57.1% | 81.3% | 80.8–57.1% | 80.8–81.3% |

Notes:
Statistically significant results were put in bold.
ZTPI, Zimbardo time perspective inventory; CT, circles test.
BPD, Borderline personality disorder; NPD, narcissistic personality disorder; other PD, other personality disorder.
[U] Mann-Whitney U.
[T] Student's T test.
[H] Kruskall-Wallis H test.

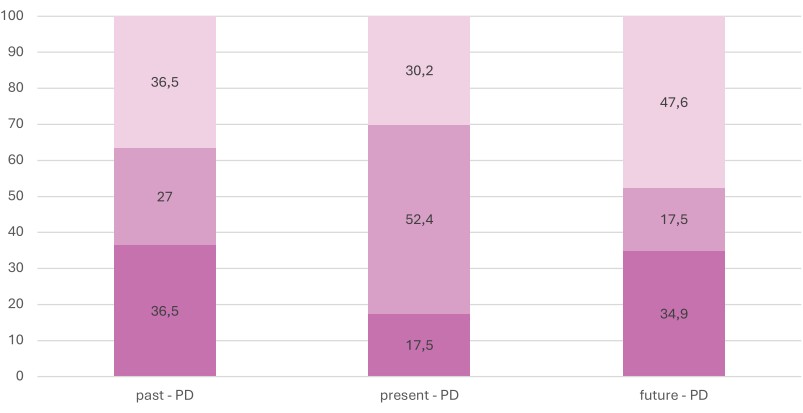

**Figure 2 Dominance of past, present and future among PD individuals.** Dark pink–dominance; medium dark pink–secondary; light pink–absence.

**Table 3 Relationship between dimensional characteristics of Personality Disorder and temporal experience (N = 63).**

| | | LPFS-BF 2.0 | | | PICD | | | | |
|---|---|---|---|---|---|---|---|---|---|
| | | Overall severity | Impairment in self-functioning | Impairment in interpersonal functioning | Negative affective | Disinhibition | Detachment | Dissocial | Anankastic |
| ZTPI | Past negative | **0.364**\*\* | **0.407**\*\* | 0.227 | **0.43**\*\* | **0.35**\*\* | 0.17 | -0.01 | −0.10 |
| | Past positive | **−0.340**\*\* | **−0.314**\* | −0.150 | −0.09 | 0.07 | −0.24 | 0.13 | 0.01 |
| | Present hedonistic | 0.020 | 0.139 | −0.027 | 0.21 | **0.68**\*\* | **−0.28**\* | **0.38**\*\* | **−0.56**\*\* |
| | Future | −0.242 | **−0.315**\* | −0.096 | −0.11 | **−0.61**\*\* | 0.23 | **−0.46**\*\* | **0.43**\*\* |
| CT | Past dominance | 0.233 | **0.445**\*\* | 0.004 | **0.254**\* | **0.256**\* | −0.022 | 0.079 | −0.209 |
| | Present dominance | 0.102 | −0.097 | 0.135 | −0.104 | −0.198 | 0.129 | 0.024 | 0.083 |
| | Future dominance | −0.130 | −0.238 | 0.099 | 0.071 | −0.078 | −0.080 | −0.103 | −0.024 |
| | Relatedness | −0.059 | −0.059 | −0.050 | −0.184 | 0.008 | −0.012 | 0.162 | 0.047 |

**Notes:**
Statistically significant results were put in bold.
ZTPI, Zimbardo time perspective inventory; CT, circles test.
\*\* Significant at 0.01 level.
\* Significant at 0.05 level.

negative past between individuals with BPD and those with NPD (U = 150.5, $p = 0.01$). Moreover, individuals with BPD were more oriented towards hedonistic present than persons with NPD (t(45) = 2.85, $p = 0.003$). The BPD uniqueness among other PD ($n = 16$) seems to lie in higher orientation on present hedonistic (H = 22.4, $p < 0.001$, df = 2) and even lower (from the already decreased in PD) future orientation (H = −14.3, $p = 0.01$, df = 2). At the same time, mean differences between NPD and other PD were insignificant, which supports BPD distinctiveness.

Following the alternative scoring of temporal relatedness proposed by Cottle, we also assessed the frequency (absolute numbers) of the atomistic perspective. The frequency of atomistic perspective was higher in BPD (80.8%) than in NPD (57.1%) and in the whole PD group (73%), but the difference was statistically insignificant ($\chi^2(2) = 4.03$, $p = 0.13$).

## Correlations between temporal experience and dimensional characteristics of PD

***Temporal experience and severity of personality dysfunction.*** The three left columns of Table 3 illustrate the relationship between the severity of personality pathology as measured by LPFS-BF 2.0 and temporal experience. More severe personality dysfunction was associated with higher levels of past negative and lower levels of past positive perspectives. Analysing the subcomponents of the overall severity score revealed that the association was derived mainly from impairment in self-functioning measures. Additionally, impairment in self-functioning had a weak, negative correlation with future perspective. A positive, moderate correlation was also observed between self-functioning impairment and past dominance as measured with CT.

***Temporal experience and maladaptive trait-domains of PD.*** The right five columns of Table 3 show the relationship between the severity of maladaptive traits and temporal experience. Among ZTPI measures, past negative perspective appeared to have moderate

and weak positive correlation with negative affective and disinhibition trait-domains, respectively. Whereas the present hedonistic perspective was negatively correlated with detachment (weak) and anankastic (moderate) and positively correlated with disinhibition (moderate) and dissocial trait-domains (weak). Additionally, future perspective was negatively correlated with disinhibition and dissocial trait-domains (both moderate), while it was positively correlated with anankastic traits (moderate). Among CT measures, past dominance exhibited a weak positive correlation with negative affective and disinhibition trait-domains.

## DISCUSSION

### Personality pathology and temporal experience in PD types

This study is the first attempt to explore the temporal experience in PD with psychometric constructs (ZTPI, CT) and interpret the findings in the context of phenomenological psychopathology. The results demonstrate that individuals with PD exhibit a heightened, reflective orientation toward past negative (PD: 4.01 ± 0.68 *vs.* healthy subjects: 2.98 ± 0.72; t(62) = −12.1, $p < 0.001$) and lowered orientation toward future (PD: 3.04 ± 0.85 *vs.* healthy subjects: 3.47 ± 0.54; t(62) = 3.99, $p < 0.001$) and past positive (PD: 2.31 ± 0.81 *vs.* healthy subjects: 3.71 ± 0.64; t(62) = 13.79, $p < 0.001$) perspectives compared to *Zimbardo & Boyd (1999)* healthy subjects ($N = 606$). At the same time, their pre-reflective experience is dominated either by the past or the future (unlike the healthy subjects directed predominantly toward the future). Based on the interval scoring, the PD group revealed significantly higher scores in past dominance (0.83 *vs.* 2.00, $p < 0.001$) and significantly lower scores in future dominance (2.87 *vs.* 1.75, $p < 0.001$) compared with Cottle's healthy sample ($N = 530$). The present dominance was not significantly different between the two groups. The passage of time was experienced atomistically more often; the mean relatedness score was lower than in *Cottle*'s *(1967)* healthy subjects (mean = 3.72). Since *Cottle (1967)* did not report SD, we could not test the statistical significance of the difference. The percentage of atomistic configuration was higher than among healthy subjects (60.8%) as calculated from *Cottle (1967)*, although the difference did not reach statistical significance ($\chi^2(1) = 1.34$, $p = 0.25$).

*Borderline Personality Disorder.* The results of this study revealed two, distinct, temporal phenotypes—of BPD and NPD. BPD has higher negative past than NPD, higher present hedonistic and lower future perspective than other PD. Time perspectives mean ± SD scores in comparison to *Mioni et al. (2020)* BPD sample ($N = 17$) and *Oyanadel & Buela-Casal (2014)* cluster B PD sample ($N = 25$) are respectively (this study Mean ± SD *vs.* *Mioni et al. (2020)* sample and *Oyanadel & Buela-Casal (2014)* sample): past negative (**4.24 ± 0.65** *vs.* 4.38 ± 0.44 and 3.83 ± 0.98), higher past positive (**3.57 ± 0.89** *vs.* 2.14 ± 0.50 and 2.93 ± 0.87), lower present hedonistic (**3.57 ± 0.89** *vs.* 3.87 ± 0.47 and 3.72 ± 0.89 ), lower future (**2.83 ± 0.67** *vs.* 3.18 ± 0.54 and 3.40 ± 0.77). Our study did not include control group, but we compared the results with reported healthy samples included in the abovementioned studies. Our participants with BPD had higher negative past (*Mioni et al., 2020*), lower positive past (*Mioni et al., 2020*; *Oyanadel & Buela-Casal, 2014*), and higher present hedonistic (*Oyanadel & Buela-Casal, 2014*) and future perspectives (*Mioni et al.,*

*2020*). It appears that BPD specificity, among other PD, lies in a unique constellation of negativity-biased past, orientation towards the hedonistic present, and diminished sense of futurity. Other PD do not share this presentistic character of experience. This may be interpreted as representing the phenomenal qualities of the "intra-festum", so feast world (*Kimura, 1992*).

*Narcissistic personality disorder.* At the same time, NPD displayed time experience profile based on: (1) lower than BPD but still higher than healthy subjects (HS) past negative; (2) similar to any other PD, but lower than HS past positive; (3) lower than BPD and HS present hedonistic; (4) comparable to BPD, but lower than HS future time perspectives. Concerning Cottle's test, both BP and NPD had higher scores in past dominance and lower in future dominance than HS. As this is the first study on a clinical sample concerning NPD temporality, the only quantitative data for comparison concerns narcissistic traits measures in a student sample, which indicates that vulnerable narcissism may be linked to higher negative past scores (*Zajenkowski et al., 2021*), while grandiose narcissism to higher present hedonism (*Zajenkowski et al., 2016*). Previous studies indicated that while in grandiose narcissism experience is more oriented toward self-bolstering future content (*Finch et al., 2024*), in vulnerable narcissism, a sense of insecurity and negative affectivity limits one's future projection (*Kealy, Sandhu & Ogrodniczuk, 2017*). However, since we did not control for narcissism subtypes, we cannot interpret our results in terms of typological variance. We may only state that they indicate that NPD partially shares the BPD temporal profile, except for the two reflective time experience measures: lower scores on present hedonistic and past negative perspectives.

The findings concerning a higher negative past and its dominance may extend previous qualitative data concerning BPD negativity bias (*Jørgensen et al., 2012*) to all PD categories. The increased past negative perspective was recently considered as constitutive of the psychopathological p-factor (*Stolarski et al., 2024*), while the diminished future perspective was also found in another study on depression (*Wang et al., 2021*). As far as the Circle Test is concerned, the only relevant mental health data concerns depression, with individuals exhibiting similar dominance of past experience, though less frequently projecting the atomistic passage of time (57.5% *vs.* 73%) (*Shigemune et al., 2021*). However, we must be wary of interpreting these quantitative overlaps as representing the same experiential phenomena. Instead, we propose to situate the results of the present study within prior phenomenological findings.

### Indirect phenomenological analysis

Previously, we hypothesized that PD experience, regardless of the subtype, revolves around cyclically experienced past scenes (*Sterna et al., 2024*). Experientially, this may be sensed as getting stuck in time, hindering one's capacity for progression and self development. Diagnostic criteria also reflect this phenomenon of a diminished capacity for future orientation in terms of low self-directedness (DSM-5, "Results"). Owing to the design of both ZTPI and Circles Test, we cannot specify the complex, felt structure of the diminished future. However, these findings may be indirectly, phenomenologically interpreted as

indicators of the future being distant, blurred, and deficient in a sense of ownership, especially when we refer to the scholarly literature on borderline personality.

One of the primary symptoms of BPD is an inability to reconcile impulses with their gratification or to foresee the consequences of actions (*Di Cesare, 2001*). BPD persons are trapped in the present moment, unable to project themselves toward the future (*Fuchs, 2007*; *Stanghellini & Mancini, 2018*). The temporal horizon remains constrained to the proximate effects of actions, which nosologically contributes to low self-directedness (*Cloninger et al., 2017*). Consequently, BPD individuals "stumble into" the future rather than "project" themselves into it (*Fuchs, 2007*, p. 381). The lived future is not so much missing as it remains restricted and shapeless, passive in its very nature. The inability to project oneself into the future, on a pre-reflective level, renders the future more of an intellectual experiment rather than an affectively powerful, meaningful force one desires to be directed toward. *Spodenkiewicz et al. (2013)* argued that the "blurred" future projection (pp. 288) is similar to the vagueness of the past, which goes in line with a bidirectional detachment of the present from the past and future.

Some BPD persons do not generate plans for the future beyond a few months, while others may construct plans that are merely declarative in nature. It seems that regardless of whether the future contains cognitive content or not, it is experienced as equally distant. According to *Rasmussen et al. (2017)*, individuals with BPD include fewer episodic details in their narratives about the future and provide "overgeneralized" (pp. 22) and nonspecific descriptions compared to individuals with ED, OCD, and a healthy control group. This aligns with the aforementioned finding that individuals with BPD score lower on the future orientation in ZTPI (*Mioni et al., 2020*). Future is akin to looking at a mountain growing in front. Cognitively, one is aware that there is likely something behind the mountain, but it is difficult to imagine its characteristics, reach it, and establish any relationship with the distant land out there (*Sterna, Moskalewicz & Fuchs, 2025*). This characteristic of lived future experience in borderline personality cannot be directly extrapolated to all the PD. Nevertheless, as we argued elsewhere, the lack of future openness is the key to inflexibility characterizing all PD (*Sterna et al., 2024*). Therefore, in the following section, we will discuss the differences and commonalities cross-categorically, incorporating a dimensional model of PD conceptualization.

## Temporal experience and dimensional characteristic of PD

**Severity.** The severity of personality pathology was associated with both reflective and projective time experience measures. The more severe impairments in self-functioning were: (1) the higher reflective, negative past perspective and pre-reflective past dominance, (2) the lower past positive and (3) future perspective.

Time experience is inevitably intertwined with self-experience, *e.g.*, in BPD sense of temporal discontinuity interplays with fragmented identity (*Fuchs, 2007*) and as such, may be initially conceptualized as comprising the phenomenological PD core. One of the crucial aspects of one's sense of self is the sense of reflexive historical continuity (*Basten & Touyz, 2020*). Moreover, the relationship between temporality and self-functioning can be

understood through subcomponents of the latter. *Skodol et al. (2011)* specify the concept of self-disturbance, introducing its three constituents: identity integration, integrity of self-concept, and self-directedness. The integrity of self-concept entails such aspects as a degree to which self-representation is complex, whereas self-directedness encloses the ability to orient oneself towards meaningful future goals. We suggest that negativity-biased, dominating past may be understood as an explicit indicator of self being implicitly reified, constrained by past scenes or traumas that remain recurringly impactful on one's functioning (*Sterna et al., 2024*). At the same time, the already discussed diminished future perspective could point to the lack of self-directedness. The interplay between lived time and self-experience described above is currently a promising hypothesis only; it should undergo rigorous validation in future studies.

**Maladaptive trait-domains.** No previous studies have examined the relationship between measures of personality disorders based on dimensional models and temporal experience. However, research indicates that the structure of personality factors in healthy individuals is continuous with the dimensions of maladaptive personality traits observed in patients with PD (*McCabe & Widiger, 2020*). In a study exploring the association between the Big-Five personality factors and temporal perspective in a normal population, several significant correlations were identified: Openness was positively correlated with a present hedonistic perspective; conscientiousness was positively correlated with a future perspective and negatively correlated with past negative and present hedonistic perspectives; extraversion was positively correlated with past positive and present hedonistic perspectives, and negatively correlated with past negative perspectives; agreeableness was positively correlated with past positive, present hedonistic, and future perspectives, and negatively correlated with past negative perspectives; and finally, neuroticism was positively correlated with past negative and future perspectives, and negatively correlated with past positive and present hedonistic perspectives (*Dunkel & Weber, 2010*).

In the present study, negative affectivity (a trait comparable to neuroticism) was positively correlated with a past negative perspective, consistent with previous findings in healthy subjects (*McCabe & Widiger, 2020*). Similarly, anankastia lies at the extreme end of conscientiousness, while impulsivity is the opposite. Therefore, the positive correlation of anankastia with a future perspective and its negative correlation with a present hedonistic perspective, as well as the reversed correlations for disinhibition, are consistent with findings from healthy subjects. Since detachment is the opposite of extraversion, the tendency for individuals with a higher detachment to have a lower present hedonistic perspective aligns with findings related to extraversion. Furthermore, the negative association between dissociality and a future perspective is consistent with the findings on agreeableness in healthy subjects.

However, some findings from our study diverged from those obtained from healthy participants. First, none of the personality traits correlated with a past positive perspective, and detachment and dissociality did not correlate with past negative perspectives. This discrepancy may be attributed to the impaired self-functioning of all participants in our

study, which likely prevented them from maintaining a positive perspective on the past, irrespective of their maladaptive personality traits.

A more interesting discrepancy is that dissociality was positively correlated with the present hedonistic perspective, even though it is considered the opposite of agreeableness. This implies that people with high present hedonistic perspective have a bimodal distribution, with one associated with high and the other with extremely low agreeableness. Similarly, among healthy individuals, neuroticism was associated with future perspective (*Dunkel & Weber, 2010*). This finding suggests that in healthy personalities, neuroticism can foster functional worry and preparation for the future. In our study, by contrast, negative affectivity was not correlated with the future perspective. This may suggest that individuals with impaired self-functioning are unable to link their anxiety and depression to constructive preparation for the future. Their perspective for the future is simply reduced.

Our study contributes novel insights into the differences in time experience among various personality disorders, employing both categorical and dimensional assessments. Specifically, individuals diagnosed with BPD exhibited a notably stronger past negative perspective compared to those with NPD and a stronger present hedonistic perspective relative to both NPD patients and others with personality disorders (PD). This was underpinned by higher levels of negative affectivity ($54.9 \pm 7.7$ *vs.* $45.9 \pm 7.6$, $t(45) = 4.0$, $p < 0.001$) and disinhibition ($39.6 \pm 6.7$ *vs.* $30.0 \pm 10.5$, $t(45) = 3.81$, $p < 0.001$), alongside lower levels of anankastia ($32.9 \pm 8.3$ *vs.* $39.6 \pm 7.8$, $t(45) = -2.84$, $p = 0.007$), when compared specifically to individuals with NPD. Moreover, compared to patients with other PD, those with BPD demonstrated higher levels of disinhibition ($39.6 \pm 6.7$ *vs.* $28.2 \pm 8.3$, $t(40) = 4.89$, $p < 0.001$) and dissociality ($28.6 \pm 7.5$ *vs.* $21.9 \pm 4.6$, $t(40) = 3.23$, $p = 0.002$), as well as lower levels of anankastia ($32.9 \pm 8.3$ *vs.* $42.2 \pm 8.3$, $t(40) = -3.52$, $p = 0.001$). These findings suggest that the heightened past negative perspective in BPD may be attributable to elevated negative affectivity, while the intensified present hedonistic perspective may stem from increased levels of disinhibition and dissociality, coupled with reduced anankastia.

## Limitations

This study has several limitations, which we elaborate upon below.

1) First, it did not collect healthy control data and used historical information, with the English version of ZTPI having a different factor structure from the Polish version; also shortened ZTPI version results were compared to the full version, so caution is advised.

2) Second, T. Cottle's projective tool has limitations; its results should be interpreted in relation to ZTPI results and previous theoretical hypotheses in order to minimize the risk of false negative or positive findings.

3) Third, the majority of patients in this study had BPD and NPD, while patients with other personality disorders were underrepresented. This sample bias stems from the fact that patients with PD outside inpatient treatment were not included. Therefore, it remains to be seen whether the results would be replicated in PD patients who attend

outpatient clinics or groups that do not seek medical care in the first place, such as those with antisocial personality disorder. Future studies on this subject should possibly compare a broader range of PD.

4) Fourth, increased past negative perspective and contraction of future perspective have been found to be not only characteristic of PD but also closely related to depressive symptoms. However, the present study did not assess scores for depressive symptoms among participants. It is not known to what extent depressive symptoms could confound the results of this study.

5) Fifth, the quantitative tools used reduced the qualitative aspects of lived temporality. For example, the Circles Test did not measure the "color" or the affective tone of temporal orientation, which usually underlies the measured dominance or absence of a temporal dimension. Future research on this subject could, therefore, include specifically phenomenological measures, such as Transdiagnostic Assessment of Temporal Experience (*Szuła, Moskalewicz & Stanghellini, 2024*), which was unavailable when we conducted this study.

6) Sixth, the study was cross-sectional and lacked any post-therapeutic follow-up. Future research should include a pre-test/post-test protocol to check to what extent the process of recovery affects the temporal profile of PD, which is very likely (*Mostowik, Mielimąka & Rutkowski, 2022*). Additionally, the applied design does not allow for causality assessment between PD and temporal experience; this could be included in future research.

## CONCLUSIONS

The results of this study extend and (to an extent) validate prior phenomenological hypotheses concerning temporality as a crucial aspect of lived experience in PD. They confirm the long-ago theorized relationship between temporal experience and self-disturbance. PD present a clear temporal profile regardless of age and duration of hospitalization, which shows that lived time qualities are independent of what happens on the clock or calendar—the objective time measures. This highlights the necessity of including temporal experience in PD lived symptomatology and possibly may necessitate targeted treatment interventions. The atomistic arrangement of temporal dimensions illustrates the broken thread of temporal becoming, a classic theme in the tradition of phenomenological psychiatry. More specifically, the overall dominance of the past and its negative weight appear critical, especially for the BPD group and those with the highest severity of PD. Combined with the diminished future perspective, these findings suggest a certain fatalistic dialectics at play—a "psychopathological spiral" to use *Melges*' *(1982)* classic term. When the self is blocked from unfolding in time towards the future, the past gains more and more weight and further constricts the temporal horizon for action. In the longer turn, it loads the present with negativity that "migrates" toward the actual calendar future, and the negative burden of the now-past increases. Additionally, given only minor differences between BPD and NPD, the findings support the dimensional approach to PD, which emphasizes the identification of lived-experience-based commonalities across

subcategories, as opposed to rigid categorization. This shift in focus may facilitate building a more comprehensive portrayal of PD experience in the future.

### Funding

Anna Sterna is a participant in the STER Internationalization of Doctoral Schools Program from NAWA Polish National Agency for Academic Exchange No. PPI/STE/2020/1/00014/DEC/02. Research was financed from large research grant from statutory funding for young researchers for 2022. Anna Sterna's and Marcin Moskalewicz's research on temporal experience was funded by the National Science Center in Poland (grant no. 2021/42/E/HS1/00106); Anna Sterna was also supported by Poznan University of Medical Sciences statutory funding for doctoral students. Eisuke Sakakibara was supported by JSPS KAKENHI (grant no. 24K15880). The funders had no role in study design, data collection and analysis, decision to publish, or preparation of the manuscript.

### Grant Disclosures

The following grant information was disclosed by the authors:
STER Internationalization of Doctoral Schools Program from NAWA Polish National Agency for Academic Exchange: PPI/STE/2020/1/00014/DEC/02.
Research was Financed from Large Research Grant from Statutory Funding for Young Researchers for 2022.
National Science Center in Poland: 2021/42/E/HS1/00106.
Poznan University of Medical Sciences.
JSPS KAKENHI: 24K15880.

### Competing Interests

The authors declare that they have no competing interests.

### Author Contributions

- Anna Sterna conceived and designed the experiments, performed the experiments, analyzed the data, prepared figures and/or tables, authored or reviewed drafts of the article, and approved the final draft.
- Eisuke Sakakibara analyzed the data, prepared figures and/or tables, authored or reviewed drafts of the article, and approved the final draft.
- Marcin Moskalewicz conceived and designed the experiments, performed the experiments, analyzed the data, prepared figures and/or tables, authored or reviewed drafts of the article, and approved the final draft.

### Human Ethics

The following information was supplied relating to ethical approvals (*i.e.*, approving body and any reference numbers):

The study protocol was approved by the hospital and Poznań University of Medical Sciences bioethical committee (decision no: KB-367/23).

### Ethics

The following information was supplied relating to ethical approvals (*i.e.*, approving body and any reference numbers):

Poznań University of Medical Science Bioethical Committee granted ethical approval to carry out the study (decision no. KB-367/23).

### Data Availability

The raw measurements are available in the Supplemental File.

### Supplemental Information

Supplemental information for this article can be found online at http://dx.doi.org/10.7717/peerj.19403#supplemental-information.

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
