# Peer review of "Time perception and lived experience in personality disorders: differences across types, dimensions and severity"

_PeerJ, doi:10.7717/peerj.19403_

## Round 0.1 · original submission · Major Revisions

Please, carefully address all the points the reviewers have indicated, especially with regards to methodology.

Reviewer 1 ·

Basic reporting

The methods, statistical analysis, and results section all have significant lack of detail (e.g., no psychometric information provided for any measures, no justification regarding which analyses were run for which test based on the shape of the distribution) and using previously published healthy control groups as the comparison group is not valid. Perhaps if the previous data were from a normed group, but even then, the previous samples were completely uncharacterized.
Further, the clinical diagnostic information is lacking here. The authors note that diagnoses were made based on ICD-10 by “skilled psychiatrists and psychologists” but do not report symptom endorsement, cutoffs, or who diagnosed the sample. Given the low reliability of PD diagnosis (e.g., Samuel, 2015), and that the manuscript--as written--hinges on a spread of personality disorder pathology representation, the diagnostic validity would need to be established before the manuscript could be published. At bare minimum, we need the interrater reliability among clinicians, the manner in which they ascertained the diagnosis (e.g., semi-structured interview, chart review, clinical consensus), and symptom level information within each diagnosis.
Additionally, the introduction reads a bit thin in terms of empirical evidence (e.g., see lines 64- 65 on p. 6). The structure of the writing implies a larger, more robust literature regarding the topic than actually exists. Leaning into what’s largely unknown would be helpful in gaining readers’ trust. Similarly, sentences such as "Lived time interplays with the level of self-disturbance and, as such, may be conceptualized phenomenologically as comprising the core of PD" border on hyperbole and are not at all supported by the presented data (although it is written beautifully).
In short, the entire paper would need significant rewrites in order to come up to publication standard.

Experimental design

Beyond uncertainty around diagnoses, the experimental design was acceptable but the authors need to choose analyses which are justified by the data at hand and which use the data at hand. The comparison to healthy groups from previously published data is invalid as written. The authors could still use their cross-sectional data to analyze their primary research questions and have a solid paper. As written, it is not up to publication standard.

Further, I would drop the circles test. Unless the authors simply failed to mention it, it has questionable validity at best and detracts from an otherwise thoughtful research question.

Validity of the findings

See comments on basic reporting and the experimental design. The findings, as written, are not valid.

·

Basic reporting

Overall, the current research study entitled “Time perception and lived experience in Personality
Disorders: differences across types, dimensions and Severity“ shows a high quality and I evaluate the investigated association between time (as a psychological construct with time perspectives and lived experience) and Personality Disorders as an innovative and important approach. There are some few but important problems with the manuscript in the current form, which needed to be addressed before the manuscript can be published.

1. BASIC REPORTING
According to the basic reporting it fulfills the formal requirements of the journal in every respect.
There is a clear unambiguous professional English language throughput all parts of the manuscript. The literature work is well done and all cited sources are relevant to the topic. There are however some aspects not addressed (see below). The structure is in line to PeerJ standards. The Figures are relevant as additional information to the information given in the text and show a high quality, and are well labelled and described. All Raw data are supplied according to the PeerJ policy.

Experimental design

2. EXPERIMENTAL DESIGN
The design of the study is not an experimental design, instead it is a cross-sectional survey among a clinical sample. As control group “historical” data were use from a larger student sample by Zimbardo and Boyd (1999). I will highlight this aspect in my summary of critical aspects of the manuscript below.
The methodology is well described and all statistical analyses were applied in excellent way.

Validity of the findings

1. VALIDITY OF THE FINDINGS
All underlying data have been provided; they are robust, statistically sound, and controlled.

Aspects which needed to adressed: I structured them by A, B, C; etc.
A. The use of short-versions oft he ZTPI is partly highly debated and critisiced in the Time Perspective literature. The corresponding literature is however ignored. See for example
Temple, E., Perry, J. L., Worrell, F. C., Zivkovic, U., Mello, Z. R., Musil, B., Cole, J. C., & McKay, M. T. (2019). The Zimbardo time perspective inventory: Time for a new strategy, not more new shortened versions. Time & Society, 28(3), 1167-1180.

Michael T. McKay, Frank C. Worrell, Elizabeth C. Temple, John L. Perry, Jon C. Cole, Zena R. Mello, Less is not always more: The case of the 36-item short form of the Zimbardo Time Perspective Inventory, Personality and Individual Differences, Volume 72,
2015, Pages 68-71,
It is necessary to give some arguments why the Polish short version is used anyway and how this can be justified. Just to give some possibilble apsects: Maybe consider short version are more accepted by clinical patients? Repsonse biases might be reduced if the survey is shorter?
B. What is not discussed in the current research is the aspect of causality, which cannot be tested by the one-time measurement of the current study with a cross-sectional design. It would however been straightforward to add a passage about potential literature into the theoretical part dealing with the potential influence of TP on PD or vice versa (see also C). Additionally, to limitation point 5.
C. What is a substantial gap in the current version of the manuscript is that the paper ignores the literature about TP-based therapeutical approaches. These should be discussed. Also, they were developed to treat PTSD, there is no reason to assume that these approaches are not relevant for personality disorder (and if so please give arguments why they are not relevant. The following literature might be helpful:

Zimbardo, P., Sword, R., & Sword, R. (2012). The time cure: Overcoming PTSD with the new psychology of time perspective therapy. John Wiley & Sons.

Zimbardo, P. G., & Sword, R. K. (2017). Living and loving better with time perspective therapy: Healing from the past, embracing the present, creating an ideal future. McFarland.

Sword, R. M., Sword, R. K., & Brunskill, S. R. (2014). Time perspective therapy: Transforming Zimbardo’s temporal theory into clinical practice. In Time perspective theory; Review, research and application: Essays in Honor of Philip G. Zimbardo (pp. 481-498). Cham: Springer International Publishing.

D. The aspect of a Balanced Time Perspective is mentioned but not addressed sufficiently in the manuscript. Much research has shown that an overall configuration is often more relevant than the influence of the single subdimensions of the ZTPI. The following literature can be helpful:

Stolarski, M., Wiberg, B., Osin, E. (2015). Assessing Temporal Harmony: The Issue of a Balanced Time Perspective. In: Stolarski, M., Fieulaine, N., van Beek, W. (eds) Time Perspective Theory; Review, Research and Application. Springer, Cham. https://doi.org/10.1007/978-3-319-07368-2_3

Stolarski, M., Bitner, J., & Zimbardo, P. G. (2011). Time perspective, emotional intelligence and discounting of delayed awards. Time & Society, 20(3), 346-363.

Maciej Stolarski, Marcin Zajenkowski, Konrad S. Jankowski, Kinga Szymaniak,
Deviation from the balanced time perspective: A systematic review of empirical relationships with psychological variables, Personality and Individual Differences,
Volume 156, 2020,

Olivera-Figueroa, L.A., Muro, A., Feliu-Soler, A. et al. The role of time perspective and mindfulness on life satisfaction in the United States of America, Spain, Poland and Japan: A cross-cultural study. Curr Psychol 42, 17682–17699 (2023).

E. In the results section of the abstract the authors report a mean-value of 7.71 for past positive of the healthy control condition. This value cannot be correct, the ZTPI sub time dimensions range from 1 to 5, so this cannot be the mean index. Cf. also Table 2 , there it seems to be correctly reported.
F. The paper compares a short version with a 56-item long version. This needed to be addressed as a limitation. Additionally, to limitation point 1.
What I wonder is why the authors did not collect additional a small sample in Poland which comparable socio-demographic features (age, income, gender). This needed to be addressed as a potential limitation.
G. The term time perception which is used in the manuscript and in in the title might be misleading. There is a lot of research dealing with subjective time duration and how this is associated to different mind sets, cognitive processes and states of consciousness. Please clarify the construct of time perspectives in the theoretical part.

Additional comments

Overall the research is very important for understanding the association between time perspective and personality disorder

---

## Round 0.2 · accepted · Accept

The authors have addressed all of the reviewers' comments.

·

Basic reporting

The authors improved thier manuscript now sufficiently.

Experimental design

The authors improved thier manuscript now sufficiently.

Validity of the findings

The authors improved thier manuscript now sufficiently.

Additional comments

The authors addressed all my concerns and improved the manuscript sufficiently.